# Model-independent determination of the cosmic growth factor

Sophía Haude[1][⋆], Shabnam Salehi[1], Sofía Vidal[1],
Matthias Bartelmann[2] and Matteo Maturi[1,2]

**1** Institute for Theoretical Astrophysics, ZAH, Heidelberg University, Germany
**2** Institute for Theoretical Physics, Heidelberg University, Germany

⋆ sophiahaude@gmx.de

## Abstract

The two most important functions describing the evolution of the universe and its structures are the expansion function $E(a)$ and the linear growth factor $D_+(a)$. It is desirable to constrain them based on a minimum of assumptions in order to avoid biases from assumed cosmological models. The expansion function has been determined in previous papers in a model-independent way using distance moduli to type-Ia supernovae and assuming only a metric theory of gravity, spatial isotropy and homogeneity. Here, we extend this analysis in three ways: (1) We enlarge the data sample by combining measurements of type-Ia supernovae with measurements of baryonic acoustic oscillations; (2) we substantially simplify and generalise our method for reconstructing the expansion function; and (3) we use the reconstructed expansion function to determine the linear growth factor of cosmic structures, equally independent of specific assumptions on an underlying cosmological model other than the usual spatial symmetries. In this approach, the present-day matter-density parameter $\Omega_{m0}$ is the only relevant parameter for an otherwise purely empirical and accurate determination of the growth factor. We further show how our method can be used to derive a possible time evolution of the dark energy as well as the growth index directly from distance measurements. Deviations from ΛCDM that we see in some of our results may be due to possibly insufficient flexibility of our method that could be cured by larger data samples, a real departure from ΛCDM at $a \lesssim 0.3$, or hidden systematics in the data. The latter could be a matter of concern for all type-Ia supernovae analyses based on ΛCDM fitting approaches, especially in view of the current dispute on the value of $H_0$. These results illustrate the applicability of our approach as a diagnostic tool.

# 1   Introduction

The expansion function of the universe and the linear growth factor of cosmic structures are the two most fundamental functions describing the evolution of the universe and its structures. They are indirectly accessible to astronomical observations, such as luminosity-distance measurements of type-Ia supernovae (SN Ia). Combining both functions allows to distinguish between different cosmological models.

The accelerated expansion rate of the universe has been established more than twenty years ago based on SN Ia distance measurements [1, 2]. The cosmological standard model explains this acceleration by the cosmological constant or a dynamical dark-energy component currently dominating the energy content of the universe [3]. The nature of the dark energy is largely unknown. So far, all attempts to derive it from fundamental theory have led to values which are way too large to explain the cosmic acceleration. Phenomenological explanations are typically based on a dark-energy equation of state, possibly varying with time. Some of them bypass fine-tuning problems, but lack fundamental justifications. Constraining the nature of the dark energy thus remains one of the most important tasks for contemporary cosmology. The two functions mentioned, the cosmic expansion function and the linear growth factor of cosmic structures, are the most important links between observations and the nature of the dark energy.

We are here proposing a method to constrain the linear growth factor of cosmic structures without reference to any specific model for the energy content of the universe. We derive the expansion function in a way similar to that proposed by [4] and [5], but in a substantially simplified and standardised manner. The only assumptions made there are that the universe is topologically simply connected, spatially homogeneous and isotropic on average, and that the expansion rate is reasonably smooth. Extending this analysis to the linear growth of cosmic structures, we only add the assumption that the linear growth of cosmic structures on the relevant scales is locally driven by Newtonian gravity.

We briefly review and revise the method of [4] in Sect. 2 and apply it exemplarily to the Pantheon sample of type-Ia supernovae (SN sample hereafter) and to the Pantheon sample combined with a sample of distance measurements from baryonic acoustic oscillations (BAO, hereafter SN-BAO sample) to obtain a purely empirical and rather tight constraint of the cosmic

expansion function. We further show how, assuming a spatially flat Friedmann-Lemaître model with dynamical dark energy, the hypothetical time evolution of dark energy can be derived from the empirically determined expansion function. In Sect. 3 we describe our method to calculate the linear growth factor, discuss the initial conditions for solving the growth equation, and present the results obtained from the SN sample and the SN-BAO sample. As an additional application we show how to derive the growth index from the expansion function. In Sect. 4, we briefly discuss the dependence of the derived functions on the data sample in view of controversies in the literature about the acquisition of existing SN samples. We illustrate the incompatibilities between two such samples by applying our method to both. Some of our derived functions deviate significantly from the ΛCDM predictions for both data samples, and we discuss possible reasons and implications. Finally, we summarise our conclusions in Sect. 5.

## 2 Cosmic expansion

### 2.1 Method

As outlined in [4], the expansion function can be deduced from the luminosity of light sources of known intrinsic luminosity, such as calibrated SNe Ia, without assuming any specific Friedmann-Lemaître model. We briefly review this method in this section in a modified, simplified, and standardised version.

Even though gravity is commonly described by general relativity (GR), we only need to assume that space-time is described by a metric theory of gravity. We thus treat space-time as a four-dimensional, differentiable manifold with a metric tensor $g$. Assuming spatial isotropy and homogeneity, this metric has to be of the Robertson-Walker form with a scale factor $a$. In general relativity, Einstein's field equations applied to the Robertson-Walker metric turn into the Friedmann equations, and the metric further specialises to the Friedmann-Lemaître-Robertson-Walker form. Then, the cosmic expansion function $E(a)$ is given in terms of the Hubble function $H(a)$ by

$$
\begin{aligned}
H^2(a) &= H_0^2 \left( \Omega_{\mathrm{r}0} a^{-4} + \Omega_{\mathrm{m}0} a^{-3} + \Omega_{\mathrm{DE}}(a) + \Omega_{\mathrm{K}0} a^{-2} \right) \\
&=: H_0^2 E^2(a) \, .
\end{aligned}
\tag{1}
$$

This defines the cosmic expansion function $E(a)$ in terms of the Hubble constant $H_0$ and the contributing energy-density parameters. These are the radiation density $\Omega_{\mathrm{r}0}$, the matter density $\Omega_{\mathrm{m}0}$, the density parameter $\Omega_{\mathrm{K}0}$ of the spatial curvature, all at the present time, and the possibly time-dependent dark-energy density parameter $\Omega_{\mathrm{DE}}(a)$. In the standard ΛCDM cosmology, $\Omega_{\mathrm{DE}}$ is replaced by the cosmological constant with the density parameter $\Omega_{\Lambda0}$ at the present time.

It is important in our context that we do *not* assume any specific parameterisation of the expansion function of the type (1). Rather, we merely assume that we can build upon an underlying, but unspecified metric theory of gravity with the two common symmetry assumptions of spatial isotropy and homogeneity. The metric must then be of Robertson-Walker form, and its single remaining degree of freedom must be described by some expansion function $E(a)$ whose form is *a priori* undetermined. We reconstruct $E(a)$ from data without adopting the parameterisation (1).

As an uncritical simplification, we further assume that the spatial sections of the space-time manifold are flat, following the empirical evidence for the spatial curvature of our Universe to be indistinguishable from zero within the limits of our observational uncertainties [6]. It would be quite straightforward to extend our analysis by replacing the radial comoving distance $w$ in Eq. (9) below by the comoving angular-diameter distance $f_K(w)$.

We modify the approach developed in [4,5] and used in [7,8] in two important ways, allowing a substantial simplification and rendering the results more portable than before. First, we use Chebyshev polynomials of the first kind $T_n(x)$, shifted to the interval $[0,1]$, as an orthonormal basis-function system (see Appendix A). Second, we do not expand the distance, but a scaled variant of the inverse expansion function $E(a)$ into these polynomials.

Given measurements of distance moduli $\mu_i$ and redshifts $z_i$, with $1 \le i \le N$, we convert the distance moduli to luminosity distances $D_{\mathrm{lum},i}$ via

$$D_{\mathrm{lum},i} = 10^{1+0.2\mu_i} \text{ pc} \tag{2}$$

and scale the redshifts $z_i$ to the variable

$$x_i := \frac{a_i - a_{\min}}{1 - a_{\min}}, \quad a_i = (1+z_i)^{-1}, \tag{3}$$

normalised to the interval $[0,1]$, where $a_{\min} = (1+z_{\max})^{-1}$ is the scale factor of the maximum redshift in the sample. We further introduce the scaled luminosity distance

$$d_i = a_{\min}^2 (1+\delta a x_i) D_{\mathrm{lum},i}, \quad \delta a := \frac{1 - a_{\min}}{a_{\min}}. \tag{4}$$

Since the uncertainties on the redshifts are very small compared to those of the distance, the relative uncertainty of $d_i$ is unchanged compared to that of $D_{\mathrm{lum},i}$. We thus obtain a scaled data sample $\{x_i, d_i\}$.

The radial comoving coordinate is

$$w(x) = \int_t^{t_0} \frac{c\, \mathrm{d}t'}{a(t')} = \int_x^1 \frac{c\, \mathrm{d}x'}{a(x')\dot{x}'} = \int_x^1 \frac{c\, \mathrm{d}x'}{a_{\min}\dot{x}'(1+\delta a x')}, \tag{5}$$

in terms of the normalised scaled factor $x$. We define

$$e(x) := [\dot{x}(1+\delta a x)]^{-1} \tag{6}$$

and use

$$\dot{x} = \frac{\dot{a}}{a_{\min}\delta a} = \frac{\dot{a}}{a}\frac{a}{a_{\min}\delta a} = H_0 E(a)\frac{1+\delta a x}{\delta a} \tag{7}$$

to write $e(x)$ as

$$e(x) = \frac{\delta a}{E(a)(1+\delta a x)^2}. \tag{8}$$

The luminosity distance in units of the Hubble radius $c/H_0$ is

$$D_{\mathrm{lum}}(x) = \frac{w(x)}{a(x)} = \frac{1}{a_{\min}^2(1+\delta a x)}\int_x^1 \mathrm{d}x'\, e(x'), \tag{9}$$

in spatially-flat geometry, using $a = a_{\min}(1+\delta a x)$. Thus, the scaled luminosity distance $d(x)$ is

$$d(x) = \int_x^1 \mathrm{d}x'\, e(x'), \tag{10}$$

and the scaled, inverse expansion function $e(x)$ is its negative derivative,

$$e(x) = -d'(x). \tag{11}$$

We now proceed as follows with the transformed data set $\{x_i, d_i\}$. We expand $e(x)$ into shifted
Chebyshev polynomials $T_n^*(x)$,

$$e(x) = \sum_{j=1}^{M} c_j T_j^*(x) \,. \tag{12}$$

Then, the scaled distances $d(x)$ are given by

$$d(x) = \sum_{j=1}^{M} c_j p_j(x) \,, \quad p_j(x) := \int_x^1 dx' T_j^*(x') \,. \tag{13}$$

Defining the matrix $P$ by its components

$$P_{ij} := p_j(x_i) \,, \quad 1 \le i \le N \,, \quad 1 \le j \le M \,, \tag{14}$$

the vector $\vec{c}$ of coefficients $c_j$ is determined by the data vector $\vec{d} = (d_i)^\top$ via

$$\vec{d} = P\vec{c} \,. \tag{15}$$

With the covariance matrix $C := \langle \vec{d} \otimes \vec{d} \rangle$ of the scaled luminosity distances $\vec{d}$, the maximum-
likelihood solution for $\vec{c}$ is

$$\vec{c} = \left(P^\top C^{-1} P\right)^{-1} \left(P^\top C^{-1}\right)\vec{d} \,. \tag{16}$$

The uncertainties $\Delta c_j$ of the coefficients and $\Delta E(a)$ of the expansion function are obtained
from the Fisher matrix $F = P^\top C^{-1} P$ in the following way. First, we diagonalise the Fisher
matrix by rotating it into its eigenframe with a rotation matrix $R$, find its eigenvalues $\sigma_i'^{-2}$
and define a vector of decorrelated coefficient uncertainties $\Delta\vec{c}' = (\sigma_1', \ldots, \sigma_M')$. Second, we
rotate this vector back into the frame of the Chebyshev polynomials and find $\Delta\vec{c} = R^\top \Delta\vec{c}'$. The
uncertainties $\Delta c_i$ obtained this way are slightly larger than the Cramer-Rao bound $F_{ii}^{-1/2}$, as
they are expected to be. Beginning with a large number $M$ of coefficients, only those are kept
which are statistically significant, i.e. which satisfy $|c_j| \ge \Delta c_j$.

## 2.2 Cosmic expansion function from the SN sample

We first reconstruct the expansion function using the Pantheon sample of type-Ia supernovae [9],
covering the scale-factor range $a \in [0.3067, 1]$. This sample has been criticised by [10] for
being significantly discrepant from another established type-Ia supernovae sample (the JLA
sample [11]) and for methods used in the post-processing of the observed data, especially
peculiar velocity corrections. We use this sample nonetheless to prove the principle. All
quantitative results in this work hinge on the reliability of the data. Possible systematics in the
data may be the most likely reason for some of our results deviating from $\Lambda$CDM.

We apply the algorithm described in the preceding subsection to derive the function $e(a)$
defined in Eq. (8). Using the covariance matrix provided with the data, we determine the
coefficient vector $\vec{c}$ using Eq. (16) and derive its uncertainty $\Delta\vec{c}$ as described above. We arrive
at $M = 3$ significant coefficients.

We then transform to $E(a)$ via Eq. (8) and determine its uncertainty from

$$\frac{\Delta E(a)}{E(a)} = \frac{\Delta e(a)}{e(a)} \,. \tag{17}$$

Our result for the expansion function and its uncertainty are shown in Fig. 1. The uncertainties
are very small because the entire information taken from the SN sample is compressed into

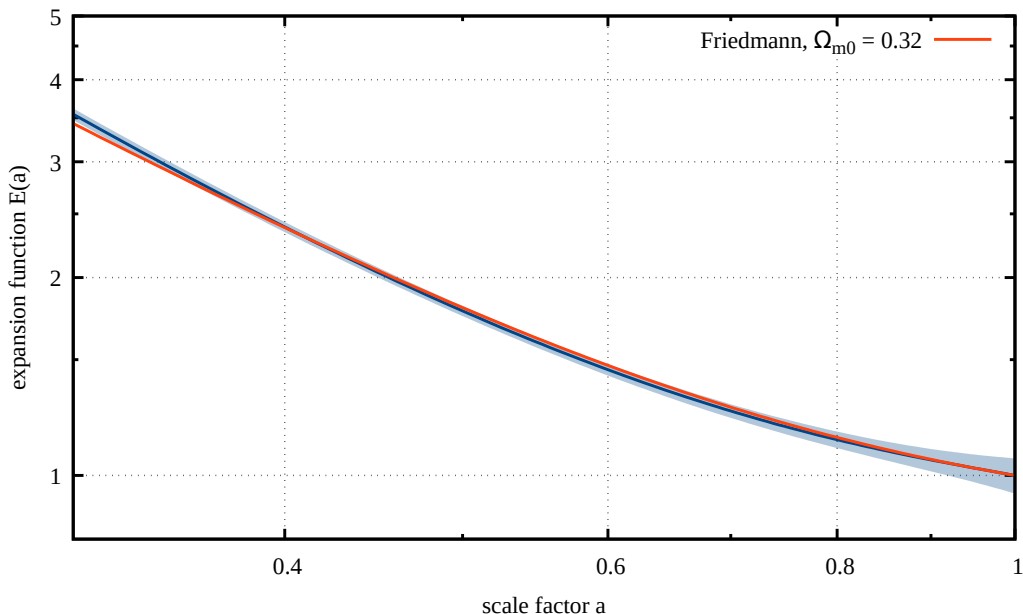

Figure 1: The cosmic expansion function $E(a)$ is shown here as reconstructed from the luminosity-distance measurements in the SN sample. Spanned by the shifted Chebyshev polynomials $T_j^*(a)$, the model needs three significant coefficients $c_j$ whose error bars are determined by the covariance matrix of the data (see the entries in Tab. 1). The 1-$\sigma$ uncertainty shown here is so small because the entire information from the data set is thus compressed into three numbers. The red line shows the best-fitting, spatially-flat, Friedmann expansion function.

three coefficients here. Another reason is that the uncertainties assigned to the Pantheon data are already very small compared to other SN samples. The best-fitting $\Lambda$CDM model with

$$E_{\Lambda\text{CDM}}(a) = \left(\Omega_{\text{m0}} a^{-3} + 1 - \Omega_{\text{m0}}\right)^{1/2} ,\qquad(18)$$

in the common parameterisation of Eq. (1) and further assuming $\Omega_{\text{r0}} = 0$ and $\Omega_{\text{K0}} = 0$, requires $\Omega_{\text{m0}} = 0.324 \pm 0.002$. It is overplotted in red in Fig. 1.

## 2.3 Cosmic expansion function from the SN-BAO sample

We repeat our analysis with the combined SN-BAO sample. We collected a sample of BAO measurements by searching the reviewed literature for papers that appeared between January, 2014, and December, 2018. We selected 21 papers according to the quality and the completeness of the data description and collected 89 measurements of the angular-diameter distance $D_{\text{ang}}/r_{\text{d,fid}}$ in terms of a fiducial value $r_{\text{d,fid}}$ for the so-called drag distance, setting the physical scale of the BAOs. The drag distance is the sound horizon at the end of the baryon-drag epoch. Of these measurements, we kept 75 after removing those that seemed to be either dependent on or superseded by other measurements (see Appendix C). These measurements fall into the redshift range $[0.24, 2.4]$ and thus extend the scale-factor range of our reconstruction of the expansion function.

The drag distance $r_{\text{d,fid}}$ is unknown to us. It is determined by

$$r_{\text{d}} = \frac{1}{H_0} \int_0^{a_{\text{d}}} \frac{c_{\text{s}}(a)\mathrm{d}a}{a^2 E(a)} \qquad(19)$$

and thus needs for its theoretical calculation the expansion function for scale factors smaller than $a_{\text{d}} \approx 1100^{-1}$. In order to remain as model-independent as possible, we choose to determine $r_{\text{d}}$

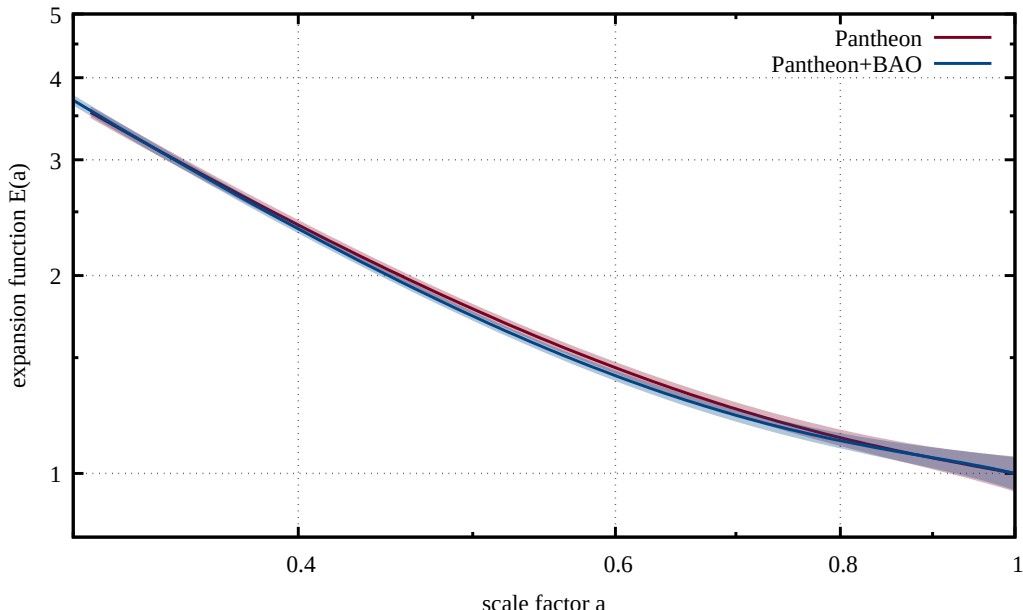

Figure 2: Expansion functions determined from the SN-BAO sample and from the SN sample alone for comparison. As in Fig. 1, 1-$\sigma$ uncertainties are shown. The best-fitting, spatially-flat Friedmann expansion function is the same as in Fig. 1 and thus not repeated here. The reconstruction of $E(a)$ from the combined samples requires four significant coefficients (cf. Tab. 1).

by an empirical calibration: we apply an offset to the distance moduli derived from the BAO measurements such as to bring them into least-squared distance with the sample of distance moduli from the SN sample. This corresponds to cross-calibrating the drag distance to match absolute SN-Ia luminosities. This offset turns out to be redshift-independent, as expected. Its value of $\Delta\mu = 10.783 \pm 0.041$ corresponds to a drag distance of

$$r_{\mathrm{d}} = 143.4 \pm 2.7\,\mathrm{Mpc}\,, \tag{20}$$

in good agreement with the value expected in the standard $\Lambda$CDM cosmology. We further estimate the covariance matrix of the BAO data via the uncertainties quoted in the papers, combine the two statistically fully independent samples and repeat the determination of the coefficients $\vec{c}$ and the expansion function as for the SN sample alone. The result is shown in Fig. 2. For the SN-BAO sample, we obtain $M = 4$ significant coefficients.

Within their uncertainties, the expansion functions obtained from the SN sample alone and from the SN-BAO sample agree with each other, but the uncertainties due to the combined sample are somewhat smaller, and the redshift range of the reconstruction is slightly extended. The fit to the standard-$\Lambda$CDM expansion function leads to a result virtually indistinguishable from the SN sample alone, with $\Omega_{\mathrm{m}0} = 0.319 \pm 0.002$, and is therefore not shown again in Fig. 2.

Interestingly, the expansion function determined purely from the data is slightly more curved than the best-fitting Friedmann-Lemaître model. This difference is formally highly significant, but, as argued above, we do not want to emphasise it since it is likely to be caused by systematic biases in the data or their interpretation. The expansion coefficients determined from both data sets, i.e. from the SN sample and from the SN-BAO sample, are listed in Tab. 1.

Albeit likely premature in view of possible systematics in the data, it is interesting to use the reconstructed expansion function to constrain the hypothetical time evolution of the dark energy. If the expansion function $E(a)$ derived from the data were to be represented by the

Table 1: Significant expansion coefficients $\vec{c}$ and their uncertainties $\Delta\vec{c}$.

| Sample | | order $j$ | | | |
|---|---|---|---|---|---|
| SN sample | $c_j$ | 0.988 | −0.372 | 0.045 | |
| | $\Delta c_j$ | 0.033 | 0.035 | 0.018 | |
| SN-BAO sample | $c_j$ | 0.983 | −0.374 | 0.034 | 0.007 |
| | $\Delta c_j$ | 0.029 | 0.032 | 0.017 | 0.001 |

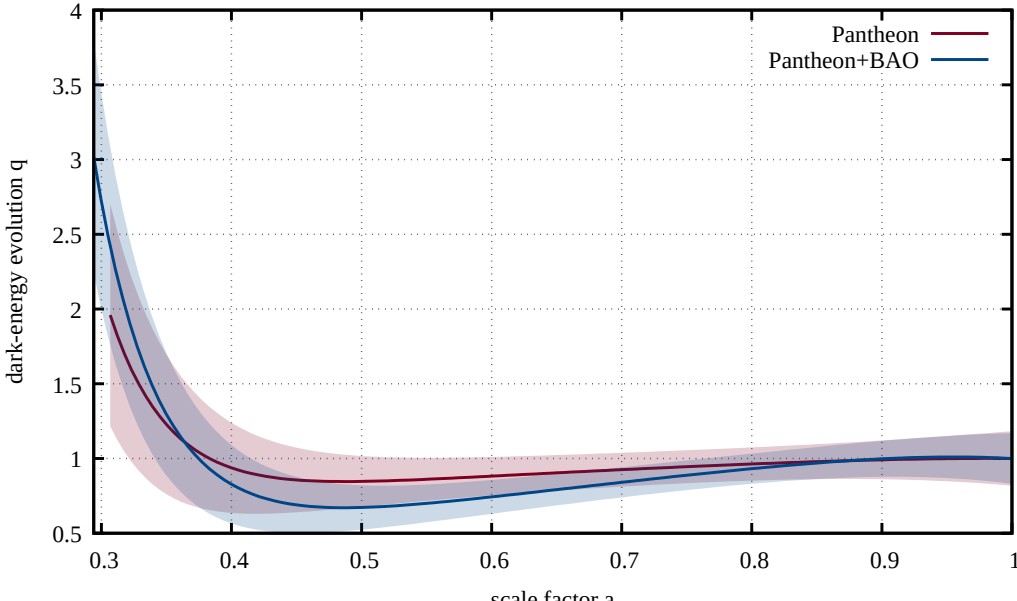

Figure 3: Constraints on a dynamical evolution of dark energy $q(a)$ as given in Eq. 22, obtained by comparing the expansion functions derived from the SN-BAO sample with the expectation for a spatially-flat Friedmann-Lemaître model (blue). The red band shows analogous constraints obtained from the SN sample only. As in Figs. 1 and 2, 1-$\sigma$ uncertainties are shown.

expansion function $E_{\Lambda\text{CDM}}(a)$ for a spatially-flat Friedmann-Lemaître model with dynamical dark energy, we should have

$$E^2(a) \overset{!}{=} \Omega_{\text{m0}} a^{-3} + (1 - \Omega_{\text{m0}}) q(a) \,, \tag{21}$$

which would imply

$$q(a) = \frac{E^2(a) - \Omega_{\text{m0}} a^{-3}}{1 - \Omega_{\text{m0}}} \,, \quad \Delta q(a) = \left| \frac{2E(a)}{1 - \Omega_{\text{m0}}} \right| \Delta E(a) \,, \tag{22}$$

for the function $q(a)$ quantifying the time evolution of the dark energy and its uncertainty. This function is shown in Fig. 3 for the SN and the SN-BAO sample, setting $\Omega_{\text{m0}} = 0.32$ as obtained from the best-fitting $\Lambda$CDM model determined above. It illustrates one of the advantages of our approach, as the empirically determined expansion function does not assume any specific cosmological model in general, nor a specific model for dynamical dark energy in particular. In view of its 1-$\sigma$ uncertainty show in Fig. 3, the derived function $q(a)$ does not deviate significantly from the $\Lambda$CDM result $q(a) = 1$.

## 3 Linear growth of cosmic structures

### 3.1 Equation to be solved

Relative to the background expanding as described by $E(a)$, structures grow under the influence of the additional gravitational field of density fluctuations $\delta\rho(\vec{x}, t) = \bar{\rho}(t)\delta(\vec{x}, t)$, where $\bar{\rho}(t)$ is the mean matter density and $\delta$ the density contrast. Structures small compared to the curvature radius of the spatial sections of the universe with a density contrast $\delta \lesssim 1$ can be treated as linear perturbations of a cosmic fluid in the framework of Newtonian gravity.

Linearising the corresponding Euler-Poisson system of equations in the perturbations and expressing spatial positions in comoving coordinates leads to the well-known second-order, linear differential equation

$$\ddot{\delta} + 2H\dot{\delta} = 4\pi G\bar{\rho}\delta \,, \tag{23}$$

for the density contrast $\delta$ of pressure-less dust. Since none of the terms in Eq. (23) depends on spatial scales, the solutions for $\delta$ can be separated into a time dependent function $D(t)$ and a spatially dependent function $f(\vec{x})$, writing $\delta(\vec{x}, t) = D(t)f(\vec{x})$, where $D(t)$ alone has to satisfy Eq. (23). Of the two linearly independent solutions of Eq. (23), one decreases with time and is thus irrelevant for our purposes. We focus on the growing solution $D_+(t)$, i.e. the linear growth factor. Transforming the independent variable in Eq. (23) from the time $t$ to the scale factor $a$ results in the equation

$$D_+'' + \left(\frac{3}{a} + \frac{E'(a)}{E(a)}\right)D_+' = \frac{3}{2}\frac{\Omega_{\mathrm{m}}}{a^2}D_+ \,, \tag{24}$$

for the linear growth factor, with primes denoting derivatives with respect to $a$. The time-dependent matter-density parameter $\Omega_{\mathrm{m}}(a)$ is given by

$$\Omega_{\mathrm{m}}(a) = \frac{\Omega_{\mathrm{m}0}}{E^2(a)a^3} \,, \tag{25}$$

in terms of the expansion function $E(a)$ and the present-day matter-density parameter $\Omega_{\mathrm{m}0}$. Equation (24) thus depends only on the expansion function $E(a)$, its first derivative, and the present matter-density parameter $\Omega_{\mathrm{m}0}$. We know $E(a)$ empirically in a model-independent way from the procedure described in Sect. 2.

### 3.2 Initial conditions and results for the linear growth factor

Before we can proceed to solve Eq. (24) for the growth factor, we need to set $\Omega_{\mathrm{m}0}$ and to specify initial conditions. Since we know $E(a)$ from data taken in the scale-factor interval $[a_{\min}, 1]$, we need to set the initial conditions at $a_{\min}$. Since Eq. (24) is homogeneous, the initial value of $D_+$ is irrelevant and can be set to any arbitrary value. We choose $D_+(a_{\min}) = 1$. Concerning the derivative $D_+'(a)$ at $a = a_{\min}$, we begin with the *ansatz* $D_+ = a^n$ near $a = a_{\min}$, assume that $n$ changes only slowly with $a$ and use Eq. (24) to find

$$n = \frac{1}{4}\left[-1 - \varepsilon + \sqrt{(1 + \varepsilon)^2 + 24(1 - \omega)}\right] \,, \tag{26}$$

for the growing solution, using the definitions

$$\varepsilon := 3 + 2\frac{\mathrm{d}\ln E}{\mathrm{d}\ln a} \quad \text{and} \quad \omega := 1 - \Omega_{\mathrm{m}}(a) \,. \tag{27}$$

In the matter-dominated phase, both $\varepsilon$ and $\omega$ are small compared to unity, and $n$ is approximately

$$n \approx 1 - \frac{\varepsilon + 3\omega}{5} \,. \tag{28}$$

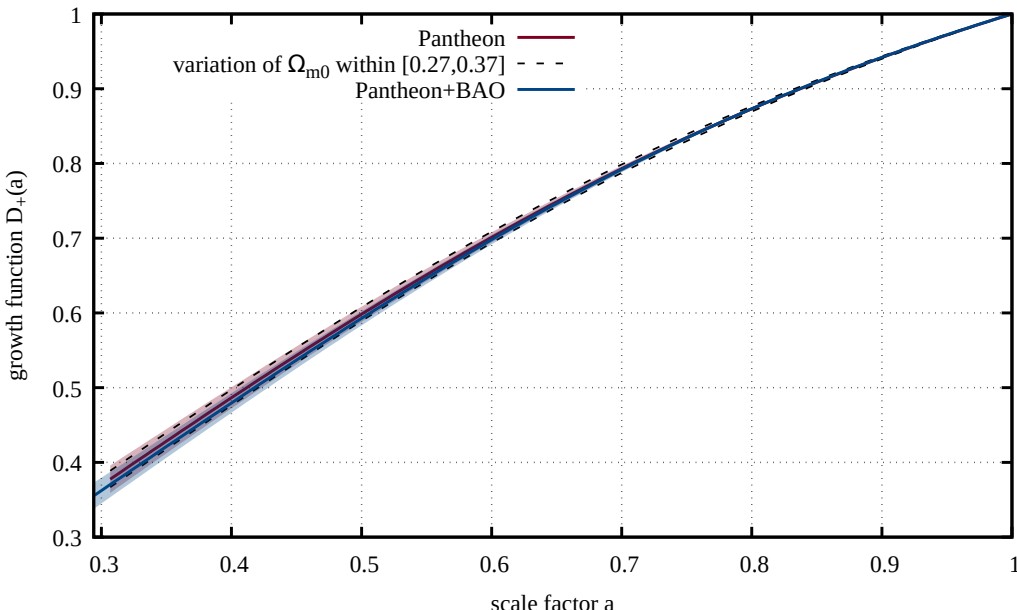

Figure 4: Linear growth factors $D_+(a)$ implied by the two expansion functions $E(a)$ shown in Fig. 2, obtained from the SN-BAO sample (blue) and from the SN sample alone (red). As described in the text, the growth factors are obtained by solving Eq. (25) using the empirically derived expansion functions and setting $\Omega_{m0}$ to the value implied by the best-fitting $\Lambda$CDM model. The shaded areas cover the 1-$\sigma$ uncertainty propagated from the uncertainty of the expansion function $E(a)$. This uncertainty is comparable to the effect of varying $\Omega_{m0}$ within $[0.27, 0.37]$ which is indicated by dashed lines in the example of the SN sample.

With the reconstructed expansion rate $E(a)$, the parameter $\varepsilon$ is fixed. For any choice of $\Omega_{m0}$, also $\omega$ is set via Eq. (25), thus so is the growth exponent $n$, and we can start integrating the growth function with the remaining initial condition

$$D'_+(a_{\min}) = n a_{\min}^{n-1} = \frac{n D_+(a_{\min})}{a_{\min}} = \frac{n}{a_{\min}} \ . \tag{29}$$

For each choice of $\Omega_{m0}$, we can now solve Eq. (24) with the initial conditions Eq. (29) and $D_+(a_{\min}) = 1$. Having arrived at $a = 1$, we normalise the growth factor such that it is unity today, $D_+(a = 1) = 1$. The uncertainty of the expansion function $E(a)$ propagates to $D_+(a)$, but the uncertainty on $D_+$ shrinks towards $a = 1$ because of this normalisation. The result is shown in Fig. 4 for $\Omega_{m0}$ as derived from the fit to the expansion function.

The uncertainties from both the growth exponent $n$ and the fitted matter-density parameter $\Omega_{m0}$, disappear in the line width of the plot. The shaded areas in Fig. 4 correspond to the propagated 1-$\sigma$ uncertainties of the coefficients $\vec{c}$ defining the expansion function and are calculated as explained in [12]. These uncertainties are comparable to the effect of varying $\Omega_{m0}$ in the range $[\Omega_{m0} - 0.05, \Omega_{m0} + 0.05]$ as indicated by the dashed lines in the same figure. Hence, the growth function depends only weakly on reasonably sized variations of $\Omega_{m0}$.

## 3.3 The growth index of linear perturbations

A common representation of the derivative of the growth factor with respect to the scale factor is given by the growth index $\gamma$, defined by

$$\frac{\mathrm{d} \ln D_+}{\mathrm{d} \ln a} =: f(\Omega_m) = \Omega_m^{\gamma(a)} \ . \tag{30}$$

Theoretically predicted values of $\gamma$ that can be found in the literature [13–20] range from approximately $\gamma = 0.4$ (for some $f(R)$ modifications of gravity [21]) to $\gamma = 0.7$. This range includes models with varying dark-energy equation of state [13,19], curved-space models [18] and models beyond general relativity [13,14,21,22]. Even for models with strongly varying $\gamma$, the values for redshifts $z \in [0,2]$ are usually very close to $\gamma \sim 0.6$.

Without further specification, Eq. (30) is obviously valid for any cosmology since the growth index $\gamma(a)$ could be any function of $a$. An advantage of writing the logarithmic slope of the growth function in this way is that $\gamma(a)$ is very well constrained for a wide range of cosmological models and can be used as a diagnostic for the classification of models based on gravity theories even beyond general relativity [13,14]. For a recent and well structured review about constraints for $\gamma$ in a wide range of models, see [14].

Another substantial advantage of Eq. (30) is that $\gamma$ happens to be almost constant within a wide range of models. [15] found a general expression for $\gamma(a)$ that applies to any model with a mixture of cold dark matter plus cosmological constant ($\Lambda$CDM) or quintessence (QCDM). For example, for a dark-energy equation of state parameterised by a slowly varying function $w(\Omega_{\mathrm{m}})$ in a spatially-flat universe, the growth index reduces to

$$\gamma = \frac{3(w-1)}{6w-5} \tag{31}$$

[16]. Thus, for any constant $w$, the growth index $\gamma$ is itself constant and reduces to $\gamma = 6/11 \approx 0.55$ for $\Lambda$CDM.

It is interesting in our context that we can derive $\gamma$ based on the reconstructed expansion function $E(a)$. As we show in Appendix B, an approximate, yet sufficiently accurate solution for $\gamma$ is

$$\gamma = \frac{\varepsilon + 3\omega}{2\varepsilon + 5\omega} \ . \tag{32}$$

For a $\Lambda$CDM model,

$$\frac{2aE'}{E} = \frac{2a}{E}\left(\frac{-3\Omega_{\mathrm{m0}}a^{-4}}{2E}\right) = -3\frac{\Omega_{\mathrm{m0}}}{E^2 a^3} = -3(1-\omega)\,, \tag{33}$$

thus $\varepsilon = 3\omega$ from Eq. (27), and Eq. (32) reduces to $\gamma = 6/11$. With our reconstruction of the expansion function $E$, we can determine $\gamma$ and its uncertainty

$$\Delta\gamma = \left[\left(\frac{\partial\gamma}{\partial c_j}\right)^2 \Delta c_j^2\right]^{1/2}\,, \tag{34}$$

for any choice of $\Omega_{\mathrm{m0}}$. The result for $\Omega_{\mathrm{m0}}$ as derived from the fit to the expansion function is shown for both data samples in Fig. 5.

The growth index follows the $\Lambda$CDM result very closely for $a \gtrsim 0.45$, but increases for smaller scale factors. Furthermore, its uncertainty for the combined SN-BAO sample is larger than for the SN sample alone. This indicates that the two data sets are not fully compatible. It is thus likely that systematic errors in the data or any unaccounted covariance between the data points is responsible for the behaviour of $\gamma$ at $a \lesssim 0.45$. We will further comment on these deviations from the $\Lambda$CDM expectation in Sec. 4. Here, we conclude by pointing out that our reconstruction method allows a direct determination of the growth index $\gamma$.

## 4 Comparison of different data samples

While our method is model-independent in the sense discussed above, existing SN and BAO samples may depend on several model assumptions because they consist of measurements that

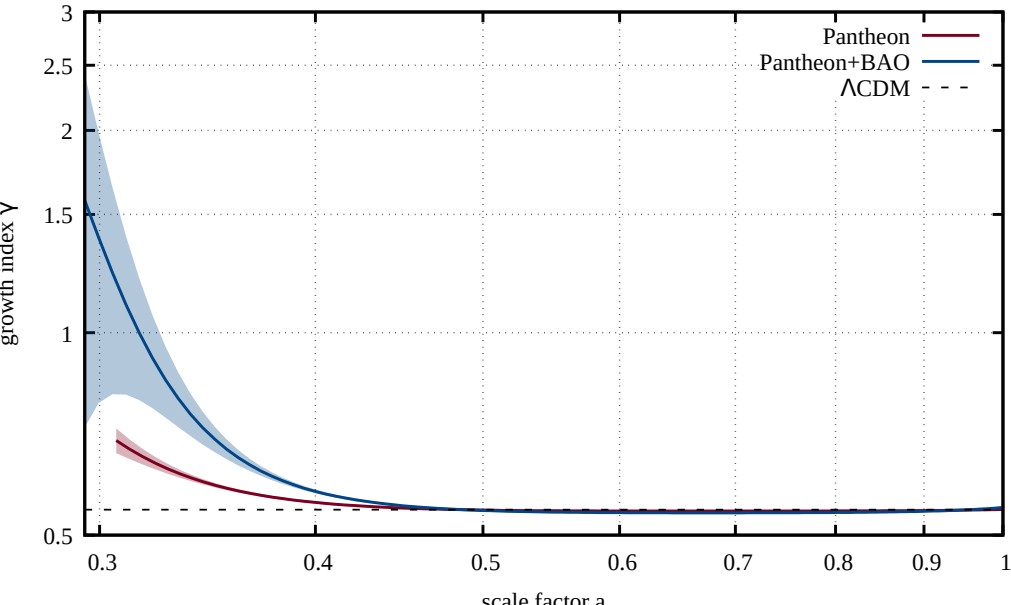

Figure 5: Growth index $\gamma$ derived from the expansion function $E$, reconstructed from the SN sample and from the SN-BAO sample, adopting $\Omega_{m0} = 0.319$. The fact that the uncertainty increases for the combined data sets at $a \lesssim 0.4$ indicates that the individual data sets are not fully compatible.

have been post-processed in sequences of non-trivial steps. Some of these steps may sensitively depend on cosmological model assumptions. Hence, while our algorithm itself makes no reference to a specific cosmological model, the functions we derive with it may reflect intrinsic model-dependences as well as biases possibly introduced into the data in the reduction process. In fact, existing data samples are partly incompatible with each other, and some post-processing steps are controversially discussed in the literature (e.g. [10]).

To give the reader an idea of how the differences between currently available SN and BAO samples affect the results obtained by our model-independent approach, we repeat the whole analysis with a second sample of type-Ia SNe (the Union-2.1 sample, [23]) alone, and with this sample combined with the BAO sample. We thus contrast results obtained from four samples, i.e. the Pantheon sample, the Union-2.1 sample, and the two combined samples Pantheon+BAO and Union-2.1+BAO.

The four functions discussed in this paper, i.e. the expansion function $E(a)$, the dark-energy evolution $q(a)$, the growth function $D_+(a)$, and the growth index $\gamma(a)$, are plotted for the four data samples in Fig. 6. This comparison is meant to illustrate variations between samples but does not allow any quantitative conclusions about the quality of individual data sets, especially since both SN samples (as well as other established samples like the JLA sample) share large amounts of raw data as well as essential post-processing tools.

The top left panel shows that the four reconstructed expansion functions are all compatible with each other in view of their uncertainties. The growth function reconstructed from the Union-2.1 sample alone is flatter than obtained from the other samples, but the three growth functions derived from the Pantheon sample alone and from the two SN+BAO samples are virtually identical. Within their 1-$\sigma$ uncertainty bounds the dark energy evolution functions $q(a)$ reconstructed from the four samples are partly not compatible with each other and with $q(a) = 1$. For scale factors $a \lesssim 0.4$, the functions derived from the Pantheon sample and the two samples combined with BAO data turn upward, but in view of their 3-$\sigma$ uncertainty bounds this tendency is so far not significant.

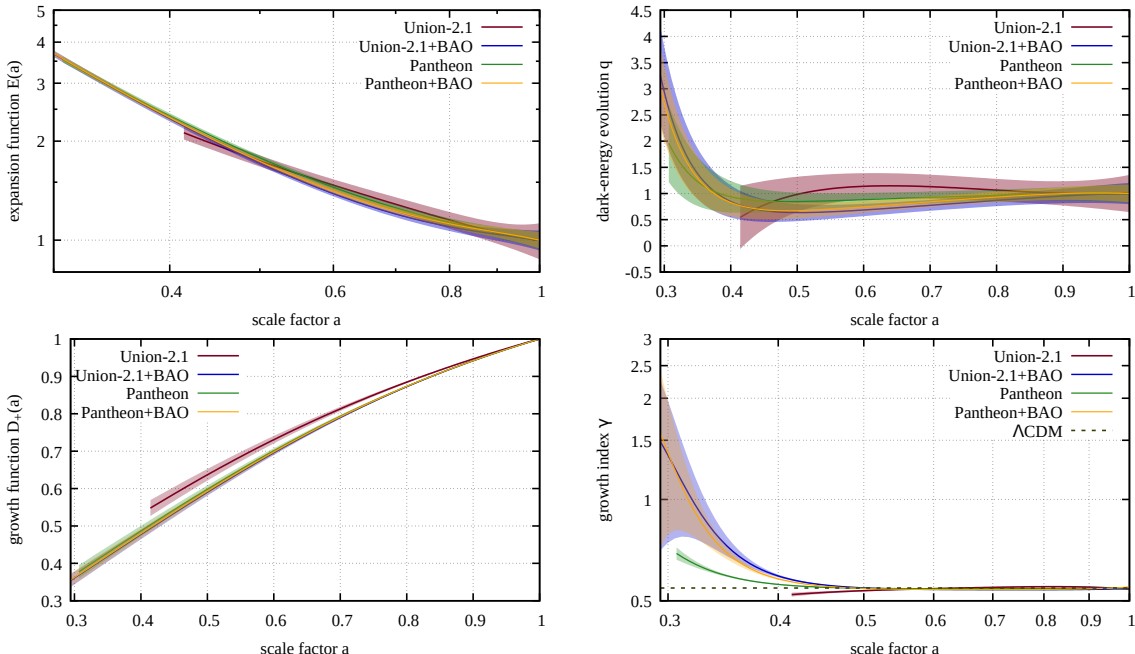

Figure 6: Each panel shows results obtained from four data samples: the Union-2.1 and Pantheon SN samples (red and green, respectively), and these two samples combined with BAOs (blue and yellow, respectively). From left to right and from top to bottom, the four panels show (a) the expansion function $E(a)$, (b) the evolution of the dark energy $q(a)$, (c) the linear growth factor $D_+(a)$, and (d) the growth index $\gamma(a)$. The growth functions obtained from the two combined SN+BAO samples are nearly indistinguishable, hence the blue line in the lower left panel is hidden behind the yellow line.

The growth index $\gamma(a)$ shows the largest variation across different samples. While it closely follows the $\Lambda$CDM expectation for all samples at scale factors $a \gtrsim 0.45$, the Pantheon and in particular that BAO data drive it to larger values for scale factors $a \lesssim 0.45$. Adding BAO data increases the uncertainty substantially, which indicates that the SN and BAO samples are to some degree incompatible with each other. Our cross-calibration of the BAO drag distance with the SNe only ensures that the BAO sample continuously extends the SN samples. This indication of partly incompatible data, and the enhanced uncertainty of the growth indices derived from the combined BAO and SN samples, leads us to the conclusion that we cannot take the apparent deviation from $\Lambda$CDM seriously yet. It might be an indication of systematics in the analyzed samples that would affect any cosmological analysis based on model fitting approaches.

Even though our method could in principle be insufficiently flexible by its restriction to the few significant Chebyshev coefficients, the expansion functions shown in Figs. 1 and 2 are rather more than less curved than the best-fitting expansion function for a spatially flat, conventional Friedmann cosmology.

We draw three main conclusions from this analysis: first, the expansion and growth functions recovered from the largest and combined samples show only little variation in view of their uncertainties; second, the empirical calibration of the drag distance is quite insensitive to differences in the SN samples and the BAO sample stabilises the results of the combined samples; and third, the variations in the growth index and its uncertainty being larger in the combined SN+BAO samples indicate that the BAO and SN data sets are not fully compatible yet.

## 5  Conclusions

We have shown here how the linear growth factor $D_+(a)$ of cosmic structures can be inferred from existing data with remarkably small uncertainty without reference to a specific cosmological model. Following up on, modifying and extending earlier studies, we have derived the cosmic expansion function $E(a)$ in a way independent of the cosmological model from the measurements of distance moduli to the type-Ia supernovae of the Pantheon sample and the Union-2.1 sample, as well as from each of the two samples combined with a sample of BAO distance measurements compiled from the literature. All we need to assume is that underlying the cosmological model is a metric theory of gravity and that our universe satisfies the symmetry assumptions of spatial homogeneity and isotropy reasonably well. The uncertainty on this empirically determined expansion function already is remarkably small, and the results obtained from the Pantheon SN sample alone and from two SN samples combined with the BAO sample agree very well with each other.

The expansion function obtained this way is the main ingredient for the differential Eq. (24) describing cosmic structure growth in the linear limit. Only one parameter is needed to solve this equation, viz. the present-day matter-density parameter $\Omega_{m0}$, because it enters into the initial conditions for solving Eq. (24) and into the equation itself. Adopting a value for $\Omega_{m0}$ derived from fitting a spatially flat Friedmann model to our reconstructed expansion function, we can also solve for the growth index $\gamma$ defined in Eq. (30). This implies that, due to measurements of the distance moduli to the type-Ia supernovae in the SN and SN-BAO samples, the expansion function is accurately determined, and the linear growth factor $D_+$ is tightly constrained up to a single remaining parameter, i.e. the present-day matter density parameter $\Omega_{m0}$.

Variations of the results mainly for the growth index $\gamma(a)$ with different data samples indicate that the data may contain systematic effects that may arise in the process of data reduction of the SN samples that could partly be caused by implicit cosmological model assumptions. Uncertainties in the growth index increasing in the combined SN+BAO samples show that the different data types do not seem to be fully compatible.

Notwithstanding their uncertainties, our results for the dark-enery evolution $q(a)$ shown in Fig. 3, and for the growth index $\gamma(a)$, shown in Fig.5, illustrate how our method can be used with future data to derive these two functions directly from distance measurements.

Some of our results, in particular the growth index and the time-dependence of the dark-energy density, deviate from the $\Lambda$CDM expectations. In the case of the dark-energy evolution function $q(a)$, these deviations are not significant. In the case of the growth index $\gamma(a)$, the most likely reasons are systematic effects in the data and partial incompatibilities between data samples, as discussed. We thus abstain from claiming any evidence for non-standard behaviour of these functions. However, the possible presence of systematics in the data samples would be a matter of concern for cosmological analyses based on $\Lambda$CDM fitting approaches, especially in view of the current dispute on the value of $H_0$.

In future work, we will extend the method presented here to further types of data. Our goal is to determine the two centrally important functions of cosmology, $E(a)$ and $D_+(a)$, with as few assumptions as possible and without reference to a specific cosmological model. Such applications of our results may be particularly interesting which so far require assuming cosmological parameters or models for a possible evolution of dark energy, e.g. cosmological weak gravitational lensing.

# Acknowledgements

It is a pleasure to thank Bettina Heinlein, Sven Meyer, Jiahan Shi, Lorenzo Speri, and Jenny Wagner for interesting and helpful discussions. This work was supported by the Deutsche Forschungsgemeinschaft (DFG) via the Transregional Collaborative Research Centre TRR 33 (MB, MM) and under Germany's Excellence Strategy EXC-2181/1 - 390900948 (the Heidelberg STRUCTURES Excellence Cluster).

## A  Chebyshev polynomials

The (unnormalised) Chebyshev polynomials of the first kind $\bar{T}_n(x)$ are defined on the interval $[-1, 1]$ by the recurrence relation

$$\bar{T}_{n+1}(x) = 2x\bar{T}_n(x) - \bar{T}_{n-1}(x) \,, \tag{A.1}$$

with $\bar{T}_0(x) = 1$ and $\bar{T}_1(x) = x$. They can be written in the form

$$\bar{T}_n(\cos\theta) = \cos n\theta \tag{A.2}$$

and are orthogonal (but not orthornomal) with respect to the weight function $w(x) = (1-x^2)^{-1/2}$,

$$\begin{aligned}
\left\langle \bar{T}_n(x)\bar{T}_m(x)\right\rangle &= \int_{-1}^{1}\frac{\mathrm{d}x}{\sqrt{1-x^2}}\bar{T}_n(x)\bar{T}_m(x) = \int_0^{\pi}\mathrm{d}\theta\,\cos n\theta\,\cos m\theta \\
&= \begin{cases} 0 & n \neq m \\ \pi & n = m = 0 \\ \pi/2 & n = m \neq 0 \end{cases} .
\end{aligned} \tag{A.3}$$

The normalised Chebyshev polynomials are thus given by

$$T_n(x) := \begin{cases} (1/\pi)^{1/2} & (n = 0) \\ (2/\pi)^{1/2}\cos(n\arccos x) & (n > 0) \end{cases} . \tag{A.4}$$

Finally, the shifted Chebyshev polynomials are defined on the interval $[0, 1]$ in terms of the Chebyshev polynomials by

$$T_n^*(x) = T_n(2x - 1) \,. \tag{A.5}$$

They are orthonormal with respect to the weight function $w^*(x) = (x - x^2)^{-1/2}$.

## B  Derivation of the growth index

In terms of the logarithmic derivative

$$f := \frac{\mathrm{d}\ln D_+}{\mathrm{d}\ln a} \tag{B.1}$$

and using the parameters $\varepsilon$ and $\omega$ introduced in Eq. (27), the linear growth equation (24) reads

$$\frac{\mathrm{d}f}{\mathrm{d}\ln a} + \frac{1}{2}(1+\varepsilon)f + f^2 = \frac{3}{2}(1-\omega) \,. \tag{B.2}$$

We write

$$\frac{\mathrm{d}f}{\mathrm{d}\ln a} = f \frac{\mathrm{d}\ln\Omega_{\mathrm{m}}}{\mathrm{d}\ln a}\frac{\mathrm{d}\ln f}{\mathrm{d}\ln\Omega_{\mathrm{m}}} \, , \tag{B.3}$$

use Eq. (25) to find

$$\frac{\mathrm{d}\ln\Omega_{\mathrm{m}}}{\mathrm{d}\ln a} = -\varepsilon \tag{B.4}$$

and Eq. (30) to write

$$\frac{\mathrm{d}\ln f}{\mathrm{d}\ln\Omega_{\mathrm{m}}} = \gamma - \omega \frac{\mathrm{d}\gamma}{\mathrm{d}\ln\Omega_{\mathrm{m}}} \, , \tag{B.5}$$

approximating $\ln\Omega_{\mathrm{m}} = \ln(1-\omega) \approx -\omega$ in the last step. Neglecting terms of order $\varepsilon\omega$, we have

$$\frac{\mathrm{d}f}{\mathrm{d}\ln a} = -\varepsilon\gamma f \, . \tag{B.6}$$

Inserting this result into Eq. (B.2), dividing by $f$ and approximating

$$f = \Omega_{\mathrm{m}}^{\gamma} = (1-\omega)^{\gamma} \approx 1 - \gamma\omega \, , \tag{B.7}$$

we arrive at

$$-\varepsilon\gamma + \frac{1}{2}(1+\varepsilon) + 1 - \gamma\omega = \frac{3}{2}\left[1 + (\gamma - 1)\omega\right] \, , \tag{B.8}$$

to linear order in $\varepsilon$ and $\omega$. Solving for $\gamma$ finally gives the result

$$\gamma = \frac{\varepsilon + 3\omega}{2\varepsilon + 5\omega} \, , \tag{B.9}$$

quoted in Eq. (32).

# C   BAO sample

The sample of BAO measurements collected from the literature is listed in Tab. 2.

Table 2: BAO data.

| $n$ | $z$ | $D_A/r_d$ | $\Delta(D_A/r_d)$ | Description | Reference |
|---|---|---|---|---|---|
| 1 | 0.240 | 5.3637 | 0.4673 | autocorrelation function of CMASS galaxies in BOSS DR12 | [24] |
| 2 | 0.240 | 5.5939 | 0.3048 | redshift-space distortion moments of LOWZ and CMASS galaxy samples in BOSS DR12 | [25] |
| 3 | 0.310 | 6.2900 | 0.1400 | tomographic configuration-space analysis of galaxy autocorrelations in BOSS DR12 | [26] |
| 4 | 0.310 | 6.2948 | 0.1963 | tomographic analysis of galaxy clustering in BOSS DR12 | [27] |
| 5 | 0.310 | 6.3045 | 0.2734 | tomographic analysis of redshift-space distortion moments in BOSS DR12 galaxies | [28] |
| 6 | 0.320 | 6.6978 | 0.2099 | autocorrelation function of CMASS galaxies in BOSS DR12 | [24] |
| 7 | 0.320 | 6.4743 | 0.1896 | redshift-space distortion moments of LOWZ and CMASS galaxy samples in BOSS DR12 | [25] |
| 8 | 0.320 | 6.6689 | 0.3943 | autocorrelation function of CMASS and LOWZ galaxies in BOSS DR12, z = 0.3-0.5 | [29] |
| 9 | 0.320 | 6.6600 | 0.1600 | analysis of redshift-space distortion moments in BOSS DR14 quasars | [30] |
| 10 | 0.360 | 7.0900 | 0.1600 | tomographic configuration-space analysis of galaxy autocorrelations in BOSS DR12 | [26] |
| 11 | 0.360 | 6.9379 | 0.2572 | tomographic analysis of galaxy clustering in BOSS DR12 | [27] |
| 12 | 0.360 | 7.0870 | 0.2390 | tomographic analysis of redshift-space distortion moments in BOSS DR12 galaxies | [28] |
| 13 | 0.370 | 7.3818 | 0.3318 | autocorrelation function of CMASS galaxies in BOSS DR12 | [24] |
| 14 | 0.370 | 6.7249 | 0.4402 | redshift-space distortion moments of LOWZ and CMASS galaxy samples in BOSS DR12 | [25] |
| 15 | 0.380 | 7.4435 | 0.2730 | galaxy clustering in BOSS DR12, combined with various priors | [31] |
| 16 | 0.380 | 7.3894 | 0.1218 | power spectrum of galaxy distribution in BOSS DR12 | [32] |
| 17 | 0.380 | 7.3894 | 0.1116 | galaxy clustering in BOSS DR12, systematic-error analysis | [33] |
| 18 | 0.400 | 7.7000 | 0.1600 | tomographic configuration-space analysis of galaxy autocorrelations in BOSS DR12 | [26] |
| 19 | 0.400 | 7.5335 | 0.2166 | tomographic analysis of galaxy clustering in BOSS DR12 | [27] |
| 20 | 0.400 | 7.6576 | 0.2407 | tomographic analysis of redshift-space distortion moments in BOSS DR12 galaxies | [28] |
| 21 | 0.440 | 8.2000 | 0.1300 | tomographic configuration-space analysis of galaxy autocorrelations in BOSS DR12 | [26] |
| 22 | 0.440 | 8.0547 | 0.1760 | tomographic analysis of galaxy clustering in BOSS DR12 | [27] |

Table 2: BAO data (continued).

| n | z | $D_A/r_d$ | $\Delta(D_A/r_d)$ | Description | Reference |
|---|---|---|---|---|---|
| 23 | 0.440 | 8.0464 | 0.1601 | tomographic analysis of redshift-space distortion moments in BOSS DR12 galaxies | [28] |
| 24 | 0.450 | 8.2881 | 0.2954 | angular galaxy clustering in SDSS DR10 | [34] |
| 25 | 0.470 | 7.7682 | 0.3869 | angular galaxy clustering in SDSS DR10 | [34] |
| 26 | 0.480 | 8.6400 | 0.1100 | tomographic configuration-space analysis of galaxy autocorrelations in BOSS DR12 | [26] |
| 27 | 0.480 | 8.6977 | 0.1895 | tomographic analysis of galaxy clustering in BOSS DR12 | [27] |
| 28 | 0.480 | 8.6059 | 0.1812 | tomographic analysis of redshift-space distortion moments in BOSS DR12 galaxies | [28] |
| 29 | 0.490 | 7.7100 | 0.3245 | angular galaxy clustering in SDSS DR10 | [34] |
| 30 | 0.490 | 8.7092 | 0.2641 | autocorrelation function of CMASS galaxies in BOSS DR12 | [24] |
| 31 | 0.490 | 8.7227 | 0.2099 | redshift-space distortion moments of LOWZ and CMASS galaxy samples in BOSS DR12 | [25] |
| 32 | 0.510 | 7.8926 | 0.2789 | angular galaxy clustering in SDSS DR10 | [34] |
| 33 | 0.510 | 8.8510 | 0.1264 | galaxy clustering in BOSS DR12, systematic-error analysis | [33] |
| 34 | 0.520 | 8.9000 | 0.1200 | tomographic configuration-space analysis of galaxy autocorrelations in BOSS DR12 | [26] |
| 35 | 0.520 | 9.0565 | 0.2031 | tomographic analysis of galaxy clustering in BOSS DR12 | [27] |
| 36 | 0.520 | 9.0465 | 0.1984 | tomographic analysis of redshift-space distortion moments in BOSS DR12 galaxies | [28] |
| 37 | 0.530 | 8.7336 | 0.6107 | angular galaxy clustering in SDSS DR10 | [34] |
| 38 | 0.550 | 8.7021 | 0.5119 | angular galaxy clustering in SDSS DR10 | [34] |
| 39 | 0.560 | 9.1600 | 0.1400 | tomographic configuration-space analysis of galaxy autocorrelations in BOSS DR12 | [26] |
| 40 | 0.560 | 9.3813 | 0.2031 | tomographic analysis of galaxy clustering in BOSS DR12 | [27] |
| 41 | 0.560 | 9.3778 | 0.2077 | tomographic analysis of redshift-space distortion moments in BOSS DR12 galaxies | [28] |
| 42 | 0.570 | 9.5241 | 0.1428 | autocorrelation function of CMASS and LOWZ galaxies in BOSS DR12, z = 0.3-0.5 | [29] |
| 43 | 0.570 | 9.4200 | 0.1300 | analysis of redshift-space distortion moments in BOSS DR14 quasars | [30] |
| 44 | 0.590 | 9.5896 | 0.1693 | autocorrelation function of CMASS galaxies in BOSS DR12 | [24] |
| 45 | 0.590 | 9.6235 | 0.1558 | redshift-space distortion moments of LOWZ and CMASS galaxy samples in BOSS DR12 | [25] |
| 46 | 0.590 | 9.4500 | 0.1700 | tomographic configuration-space analysis of galaxy autocorrelations in BOSS DR12 | [26] |
| 47 | 0.590 | 9.5167 | 0.2301 | tomographic analysis of galaxy clustering in BOSS DR12 | [27] |
| 48 | 0.590 | 9.6347 | 0.2279 | tomographic analysis of redshift-space distortion moments in BOSS DR12 galaxies | [28] |

Table 2: BAO data (continued).

| $n$ | $z$ | $D_{\mathrm{A}}/r_{\mathrm{d}}$ | $\Delta(D_{\mathrm{A}}/r_{\mathrm{d}})$ | Description | Reference |
|---|---|---|---|---|---|
| 49 | 0.610 | 9.6292 | 0.1593 | galaxy clustering in BOSS DR12, systematic-error analysis | [33] |
| 50 | 0.640 | 9.9011 | 0.2844 | autocorrelation function of CMASS galaxies in BOSS DR12 | [24] |
| 51 | 0.640 | 9.7792 | 0.2777 | redshift-space distortion moments of LOWZ and CMASS galaxy samples in BOSS DR12 | [25] |
| 52 | 0.640 | 9.6200 | 0.2200 | tomographic configuration-space analysis of galaxy autocorrelations in BOSS DR12 | [26] |
| 53 | 0.640 | 9.5573 | 0.2775 | tomographic analysis of galaxy clustering in BOSS DR12 | [27] |
| 54 | 0.640 | 9.8065 | 0.3849 | tomographic analysis of redshift-space distortion moments in BOSS DR12 galaxies | [28] |
| 55 | 0.800 | 10.3720 | 0.9699 | Fourier-space measurement of clustering of eBOSS DR14 quasars | [35] |
| 56 | 0.800 | 10.8119 | 1.1428 | clustering of 147000 eBOSS DR14 quasars | [36] |
| 57 | 0.978 | 10.7334 | 1.9281 | tomographic analysis of quasar clustering in eBOSS DR14 | [37] |
| 58 | 1.000 | 12.0449 | 0.9880 | Fourier-space measurement of clustering of eBOSS DR14 quasars | [35] |
| 59 | 1.000 | 11.5205 | 1.0319 | clustering of 147000 eBOSS DR14 quasars | [36] |
| 60 | 1.230 | 11.9710 | 1.0805 | tomographic analysis of quasar clustering in eBOSS DR14 | [37] |
| 61 | 1.500 | 12.0693 | 0.7443 | Fourier-space measurement of clustering of eBOSS DR14 quasars | [35] |
| 62 | 1.500 | 12.1559 | 0.7362 | clustering of 147000 eBOSS DR14 quasars | [36] |
| 63 | 1.520 | 12.5186 | 0.7443 | combination of power spectrum and bispectrum of BOSS DR12 galaxies | [38] |
| 64 | 1.520 | 12.5186 | 0.6767 | clustering of 148659 quasars from eBOSS DR14 survey | [39] |
| 65 | 1.526 | 11.9689 | 0.6536 | tomographic analysis of quasar clustering in eBOSS DR14 | [37] |
| 66 | 1.944 | 12.2343 | 0.9911 | tomographic analysis of quasar clustering in eBOSS DR14 | [37] |
| 67 | 2.000 | 12.3585 | 0.5391 | Fourier-space measurement of clustering of eBOSS DR14 quasars | [35] |
| 68 | 2.000 | 12.0111 | 0.5616 | clustering of 147000 eBOSS DR14 quasars | [36] |
| 69 | 2.200 | 12.1697 | 0.4969 | Fourier-space measurement of clustering of eBOSS DR14 quasars | [35] |
| 70 | 2.200 | 11.8546 | 0.5392 | clustering of 147000 eBOSS DR14 quasars | [36] |
| 71 | 2.225 | 10.0425 | 1.7588 | autocorrelation function of BOSS DR12 quasars | [40] |
| 72 | 2.330 | 11.3423 | 0.6396 | Lya forest in 157783 BOSS DR12 quasars | [41] |
| 73 | 2.340 | 11.2754 | 0.6513 | Lya forest in 137562 BOSS DR11 quasars | [42] |
| 74 | 2.360 | 10.8000 | 0.4000 | Lya forest in 137562 BOSS DR11 quasars | [42] |
| 75 | 2.400 | 10.5000 | 1.2513 | cross-correlation between 234367 quasars and 168889 forests in BOSS | [43] |

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
