# Peer review of "Model-Independent Determination of the Cosmic Growth Factor"

_SciPost Astronomy, doi:SciPost Astro. 2, 001 (2022)_

## Round 1 · Referee Report · Anonymous · 2021-7-2

Report
Dear Authors,
thanks for addressing my previous report. I believe that the paper can be accepted to publication in it's present form.
Anonymous on 2021-06-24 [id 1520]
This is a comment by the authors: Due to a mistake in plain-text export, our response to the referee is somewhat garbled. We repeat it as intended here.
--
Dear Editors, dear Referee,
We would like to thank the referee for her/his constructive feedback on our paper "Model-Independent Determination of the Cosmic Growth Factor", for her/his time and effort, and for the valuable suggestions. With a long delay, for which we apologize, we have considered all of them and updated our manuscript accordingly. In the following, we will explain for each point individually to what extent and in what way we have addressed it in the revised manuscript. The comments of the referee are marked and indented, and our answers follow below, respectively.
We hope that our manuscript is now acceptable for publication in SciPost.
Kind regards,
Sophia Haude (on behalf of the authors)
--
List of changes:
Referee:
Reply:
The main goal of our paper is to provide a method that empirically reconstructs cosmological functions independent of specific cosmological models but based on observational data. Of course, as in any data-based analysis, the quality of the results depends on the quality of the data. In the original version of this paper our focus was on the proof of principle. Nevertheless, we agree with the referee's criticism that, firstly, this should be expressed more clearly and that, secondly, a discussion of the validity of the method as applied to existing data was missing although it is of particular interest. We accounted for these two points in the following way:
We added a paragraph in section 2.2 briefly pointing out the controversy about the Pantheon sample.
We refer to the resulting additional uncertainty at several points in the text where uncertainty considerations are involved.
We added a section 'Comparison of different data samples', showing all functions (expansion function, growth factor, growth index, and dark energy evolution) as derived from two different data samples (Pantheon and Union2.1 which we had not considered in the original version). We extensively discuss how the four functions react to the different samples, what this implies and what it does not imply.
Please note that this comparison does not allow any quantitative conclusions about the quality of the individual data samples, especially since both samples (as well as the JLA sample used for comparison by Rameez and Sarkar) share large amounts of raw data as well as essential post-processing tools.
We would like to emphasize that when combining the supernova samples with BAO data, the dependence of all functions on the respective supernova sample is substantially weakened.
Referee:
Reply:
Eq. 18 is indeed a simplified version of Eq. 1 where radiation and curvature terms are neglected. It is used solely to determine the ΛCDM present-day matter-density Ω_m0 (given the model-independent expansion function) and is not used for the determination of E(a). The value of Ω_m0 depends on this simplification and is thus model-dependent. This is why we state that Ω_m0 is the "only relevant parameter for an otherwise purely empirical determination" of the functions which follow. To sum this up: the expansion function is not characterized by Ω_m0, whereas the other functions are characterized by Ω_m0 only. The text now extensively discusses these points.
Referee:
As suggested by the reviewer, we implemented all these changes.

---

## Round 1 · Author Response

dear Referee,
We would like to thank the referee for hiser/heris constructive feedback on our paper “Model-Independent Determination of the Cosmic Growth Factor,“, for hiser/heris time and effort, and for the valuable suggestions. With a long delay, for which we apologize, we have considered all of them and updated our manuscript accordingly. In the following, we will explain for each point individually to what extent and in what way we have addressed it in the revised manuscript. The comments of the referee are printed in italics and indented, and our answers follow below, respectively.
We hope that our manuscript is now acceptable for publication in SciPost.
Kind regards,
Sophia Haude (on behalf of the authors)

---

## Round 1 · List of Changes

Referee:
1. Pantheon sample has been criticized by M. Rameez et al in arxiv:1911.06456. I recommend the authors to address and/or account for points presented in this paper.
Reply:
The main goal of our paper is to provide a method that empirically reconstructs cosmological functions independent of specific cosmological models but based on observational data. HenceOf course, as in any data-based analysis, the quality of the results depends on the quality of the data. If they contain errors, so do our numbers - but the method remains valid. We were less concerned with the specific numbers than withn the original version of this paper our focus was on the proof of principle.
Nevertheless, we agree with the referee‘s criticism that, firstly, this should be expressed more clearly and that, secondly, a discussion of the validity of the method as applied to existing data iswas missing although it is of particular interest.
We accounted for these two points in the following way:
1) We added a paragraph in section 2.2 briefly pointing out the controversy about the Pantheon sample.
2) We refer to the resulting additional uncertainty at several points in the text where uncertainty considerations are involved.
3) We added a section ‘Variations of the data sampleComparison of different data samples,’, showing all functions (expansion function, growth factor, growth index, and dark energy evolution) as derived from two different data samples (Union2.1 and Pantheon and Union2.1 which we had not considered in the original version). The first two functions are not varying so much butWe extensively discuss how the latter two the four functions react quite sensitively to the different samples., what this implies and what it does not imply.
NPlease note that this comparison, albeit interesting, does not allow any quantitative conclusions about the quality of the individual data samples, especially since both samples (as well as the JLA sample used for comparison by Rameez and Sarkar) share large amounts of raw data as well as essential post-processing tools.
However, also note that adding the BAO sample to both samples weakens the dependence of all functions on the supernova sample significantly, although the BAO samples are calibrated with the SN samples.
We would like to emphasize that when combining the supernova samples with BAO data, the dependence of all functions on the respective supernova sample is substantially weakened.
Referee:
2. Eq. 18 looks for me as a simplified form of Eq. 1 in which radiation and curvature terms are neglected. If this is the case it is trivial that the expansion function and any derived quantities are characterized by a matter density parameter only; I recommend authors to clarify this point.
I am not sure that in this case the uncertainty on Ωm0 has physical meaning. It is formally correct, but seems to rely on neglecting of radiation/curvature densities which are by an order of magnitude comparable to the reported uncertainty.
Reply:
Eq. 18 is indeed a simplified version of Eq. 1 where radiation and curvature terms are neglected. However, this simplification is not used at all to determine the expansion function, but only to determine the ΛCDM present-day matter-density Ωm0 given the expansion function which was determined in a model-independent way before. ‘Model-independent’ means that the specific parametrisation given in Eq. 1 has not been used for the determination of E(a)It is used solely to determine the ΛCDM present-day matter-density Ωm0 (given the model-independent expansion function) and is not used for the determination of E(a). The value of Ωm0 depends on this simplification and is thus model-dependent. This is why we state that Ωm0 is the “only relevant parameter for an otherwise purely empirical determination” of the functions which follow. To sum this up: the expansion function is not characterized by Ωm0 , whereas the other functions are characterized by Ωm0 only. This is not trivial, since they would usually at least depend on everything the expansion function is depending on including parameterisations of ΩDE.
Again, we agree that all this could be made clearer. We try to achieve this by means of numerous changes in the formulations.The text now extensively discusses these points.
Referee:
3. I suggest authors to consider replace Fig.1-2 with log-log scale versions to make uncertainty region a bit more clear. Please also refer in Fig.3 to eq. 22 which provides the definition of "dark energy evolution". Please change also y-label in this Fig. A accordingly.
Reply:
As suggested by the reviewer, we implemented all these changes.

---

## Editorial Decision

published